# COOT: Cooperative Hierarchical Transformer for Video-Text Representation Learning

**Simon Ging**[1*], **Mohammadreza Zolfaghari**[1*], **Hamed Pirsiavash**[2], **Thomas Brox**[1]

[1]University of Freiburg, [2]University of Maryland Baltimore County

[1]{gings, zolfagha, brox}@cs.uni-freiburg.de, [2] hpirsiav@umbc.edu

## Abstract

Many real-world video-text tasks involve different levels of granularity, such as frames and words, clip and sentences or videos and paragraphs, each with distinct semantics. In this paper, we propose a **Coo**perative hierarchical **T**ransformer (**COOT**) to leverage this hierarchy information and model the interactions between different levels of granularity and different modalities. The method consists of three major components: an attention-aware feature aggregation layer, which leverages the local temporal context (intra-level, e.g., within a clip), a contextual transformer to learn the interactions between low-level and high-level semantics (inter-level, e.g. clip-video, sentence-paragraph), and a cross-modal cycle-consistency loss to connect video and text. The resulting method compares favorably to the state of the art on several benchmarks while having few parameters. All code is available open-source at https://github.com/gingsi/coot-videotext

## 1 Introduction

Representation learning based on both vision and language has many potential benefits: visual grounding[1, 2, 3, 4]; visual learning with a more natural, almost self-supervised annotation process; and direct applicability to cross-modal tasks, such as video retrieval by text[5, 6, 7, 8, 9], video summarization [10], and automated indexing. This research direction has recently boomed [11, 12, 13, 14, 15, 8, 16, 17] also due to the success of self-attention in text analysis [18, 19] with its almost immediate applicability in the cross-modal context. Many different research foci are currently developing in this area, where some are concerned with large-scale pretraining to leverage the abundant data available [17, 8, 16, 11] to learn a joint embedding space, and others to bring in more explicit structure [20, 21, 22] or new losses [17, 23] into the learning process.

In this paper, we focus on long-range temporal dependencies and propose a hierarchical model that can exploit long-range temporal context both in videos and text when learning the joint cross-modal embedding. For instance, the action of "making tea" involves boiling water, pouring it into a cup, and then adding a tea bag. This action can take a long time and may have lots of details that distinguish a particular style of making tea from other styles. To capture the whole temporal context, we leverage the idea of a hierarchical model with losses that enforce the interaction within and between different hierarchy levels. The idea of such a hierarchy is generic and has been explored by several works [21, 24, 22] in the context of video-text learning. In addition, we use alignment losses from Zhang et al. [21] and extend our baseline model with a new feature aggregation method for the intra-level interactions between features and a new transformer-based module for inter-level interactions (between local and global semantics). We consider three different levels of hierarchy: frame/word, clip/sentence and video/paragraph, visualized by the three blocks in Figure 1.

---

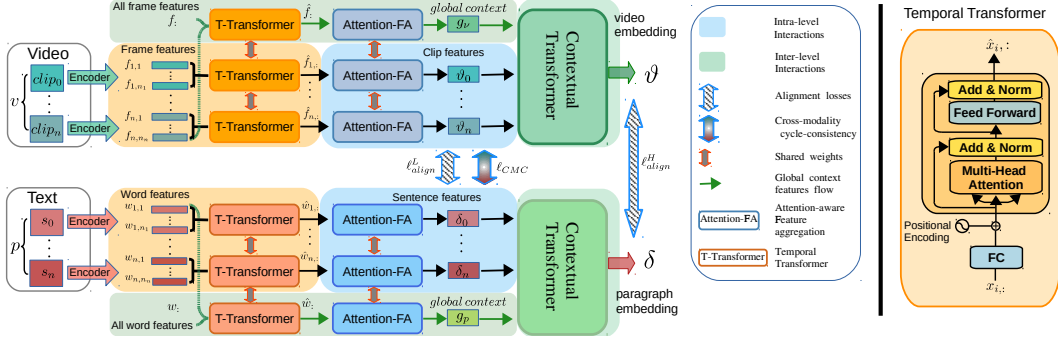

Figure 1: **Overview of COOT model** (best viewed in color). The model consist of two branches: one for video input (top) and one for text input (bottom). Given a video and a corresponding text, we encode them to frame-level/word-level features. Features belonging to each segment (clip/sentence) are fed to a standard temporal transformer (T-Transformer) followed by the proposed feature aggregation module (Attention-FA) to obtain clip/sentence-level features. Finally, a new contextual transformer produces the final video/paragraph embedding based on interactions between local context (clip/sentence features) and global context (all frames/words features). $\ell_{align}^L$, $\ell_{align}^H$, $\ell_{align}^g$ and $\ell_{CMC}$ enforce the model to align the representations at different levels.

To model *intra-level cooperation*, we introduce an attention-aware feature aggregation layer to focus on temporal interactions between low-level entities (Figure 1-Attention-FA).

This component replaces traditional sequence representation aggregation methods in transformers such as using a `[CLS]` token [19, 11, 14, 15] or mean pooling [25] with an attention-aware fusion. It leverages temporal context to encourage important entities to contribute more to the final representation of a sequence of frames or words.

For the *inter-level cooperation*, we introduce a contextual attention module, which enforces the network to highlight semantics relevant to the general context of the video and to suppress the irrelevant semantics. This is done by modeling the interaction between low-level (clips-sentences) and high-level entities (global contexts), as shown in Figure 1-green region.

In addition to this architectural contributions, we introduce a new *cross-modal cycle-consistency loss* to enforce interaction between modalities and encourage the semantic alignment between them in the learned common space. We show that enforcing two domains to produce consistent representations leads to substantially improved semantic alignment.

In summary, this paper contributes:

- a hierarchical transformer architecture with a new attention-aware feature aggregation layer and a new contextual attention module;
- a cross-modal cycle-consistency loss that encourages semantic alignment between vision and text features in the joint embedding space;
- state-of-the-art results on video-text retrieval.

## 2    Cooperative Hierarchical Transformer

Videos and text descriptions naturally involve different levels of granularity. Every paragraph contains multiple sentences, and each sentence is composed of several words. Similarly, videos have a hierarchical semantic structure, even if it is not as exactly defined as for text. Figure 1 illustrates the overview of the COOT model which consists of three levels: 1) A temporal transformer that captures the relationships between frame/word features, 2) attention-aware feature aggregation to produce clip/sentence features (Sec. 2.2), and 3) a contextual transformer to produce final video and text embeddings (Sec. 2.3). We use alignment losses from Zhang et al. [21] to align representations at different granularity levels. In addition, we introduce a new cross-model cycle-consistency loss to connect video and text (Sec. 3). In this section, we briefly summarize the alignment losses from Zhang et al. [21] and the standard transformer.

## 2.1 Preliminaries

**Semantic Alignment Losses.** For the video-text alignment, Zhang et al. [21] leverage a contrastive loss to enforce the positive samples to stay in a close neighborhood and negative samples far apart [26, 27, 28, 29]. Assuming the positive pair $\mathcal{P} = (x, y)$, two negative pairs $(x, y')$ and $(x', y)$ expressed as $\mathcal{N} = \{(x, y'), (x', y)\}$, and a margin $\alpha$, they define the following loss:

$$L(\mathcal{P}, \mathcal{N}, \alpha) = max(0, \alpha + D(x, y) - D(x', y)) + max(0, \alpha + D(x, y) - D(x, y')) \quad (1)$$

where $D(x, y) = 1 - x^\intercal y / (\|x\| \|y\|)$ is the cosine distance of two vectors.

To align representations at clip-sentence $(\vartheta_i^k, \delta_i^k)$, video-paragraph $(\vartheta^k, \delta^k)$ and global context $(g_v, g_p)$ levels, Zhang et al. [21] use the following losses:

$$
\begin{aligned}
\ell_{align}^L &= \sum_{k \in \mathcal{D}, i, k' \neq k, i' \neq i} L((\vartheta_i^k, \delta_i^k), \{(\vartheta_i^k, \delta_{i'}^{k'}), (\vartheta_{i'}^{k'}, \delta_i^k)\}, \beta) \\
\ell_{align}^H &= \sum_{k \in \mathcal{D}, k' \neq k} L((\vartheta^k, \delta^k), \{(\vartheta^k, \delta^{k'}), (\vartheta^{k'}, \delta^k)\}, \alpha) \\
\ell_{align}^g &= \sum_{k \in \mathcal{D}, k' \neq k} L((g_v^k, g_p^k), \{(g_v^k, g_p^{k'}), (g_v^{k'}, g_p^k)\}, \alpha_g)
\end{aligned}
\quad (2)
$$

Here, $\vartheta_i^k$ denotes the embedding for the $i$-th clip of the $k$-th video and similarly $\delta_i^k$ is the embedding of the $i$-th sentence of the $k$-th paragraph. $\alpha$, $\alpha_g$ and $\beta$ are constant margins, and $\mathcal{D}$ is a dataset of videos with corresponding text descriptions. Zhang et al. [21] employed an additional loss to model the clustering of low-level and high-level semantics in the joint embedding space:

$$
\begin{aligned}
\ell_{cluster} = &\sum_{k \in \mathcal{D}, i, k' \neq k, i' \neq i} L((1, 1), \{(\vartheta_i^k, \vartheta_{i'}^{k'}), (\delta_{i'}^{k'}, \delta_i^k)\}, \gamma) \\
&+ \sum_{k \in \mathcal{D}, k' \neq k} L((1, 1), \{(\vartheta^k, \vartheta^{k'}), (\delta^{k'}, \delta^k)\}, \eta)
\end{aligned}
\quad (3)
$$

where $\gamma$ and $\eta$ both are constant margins. The $(1, 1)$ pairs denote that positive samples are not changed. In short, the goal of this loss is to push apart embeddings for negative samples.

**Note.** Due to the symmetrical design of the video and text branches in our model, from now on, we explain only the video branch. For simplicity, we assume a single head in transformer formulations. All transformers use residual connections.

**Temporal Transformer** We use standard attention-blocks [18] to learn frame and word representations, as shown in Fig 1-Right. We learn two temporal transformers (T-Transformer); one for the video branch and another one for the text branch. Both have the same architecture. All T-Transformers in each branch share their weights. This module draws the relationship between temporal features and yields improved representations as output. Given a video $v^k$, we first encode all its frames to obtain the frame-level features $\{f_{i,:}^k\}_{i=1}^n$, where $f_{i,:}^k$ are all frame-level features of the $i$-th clip for video $v^k$ (orange parts in Figure 1). We also consider all frame features $(f_:^k)$ of a video as extra input for the global context computation (green parts in Figure 1). This yields $\{\hat{f}_{i,:}^k\}_{i=1}^n$ and $\hat{f}_:^k$.

## 2.2 Intra-Level Cooperation

Standard feature fusion methods consider each feature independently by average pooling or max pooling. Hence, they miss the relationship between features to highlight the relevant features. Recent transformers use a `[CLS]` token [11, 19, 14, 15] or average pooling [25] to obtain the aggregated features. For example, when a person is cooking, objects on the table are more relevant than objects on the wall or in the background. Therefore, we need to attend to specific features depending on the context. There have been some attempts in other domains to design a context-aware feature fusion method [30, 31, 32, 33]. However, we introduce an attention-aware feature aggregation module (Attention-FA in Fig. 1) for video-text transformers.

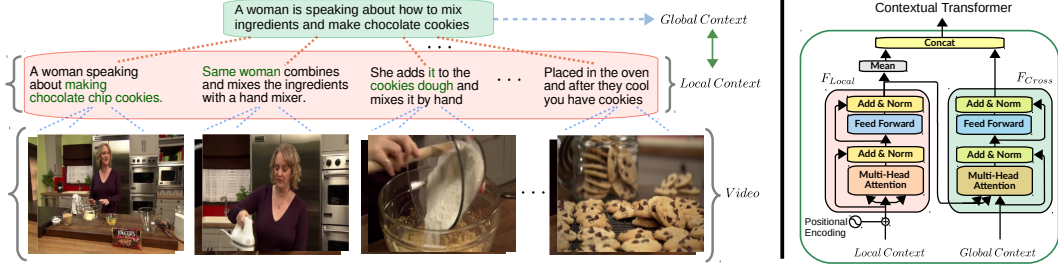

Figure 2: **Contextual Transformer (CoT).** This module (right) encourages the model to optimize the representations with respect to interactions between local and global context. In the third sentence, to know the type of dough (*cookie*) the model should have information about the general context of the video (*making chocolate cookies*). Likewise, in the second sentence, to know that she is the *"same woman"*, the model must be aware of the person's identity throughout the video.

Suppose we have a sequence with $T$ feature vectors, denoted by $X = \{x_1, \ldots, x_T\}$ (e.g. $\hat{f}_{i,:}^k = \{\hat{f}_{i,1}^k, \ldots, \hat{f}_{i,T}^k\}$). We set **key** $K = X$ and utilize two learnable transformation weights $W_2$ and $W_1$ together with two biases $b_1$ and $b_2$. The attention matrix $A$ is computed as:

$$A = \text{softmax}(W_2 Q + b_2)^T, \qquad Q = GELU(W_1 K^T + b_1), \; K = X \tag{4}$$

We compute the final feature as $\hat{x} = \sum_{i=1}^{T} a_i \odot x_i$, where $\odot$ denotes element-wise multiplication and $a_i$ is the $i$-th attention vector of $A$ for the $i$-th feature. This module differs from attention [18] in two aspects: (1) we use only two learnable weights for query ($Q$) and key ($K$) and then aggregate the values based on calculated scores; (2) the query equals to transformed keys (K) and then we apply the activation function GELU [34, 35]. We feed $\{\hat{f}_{i,:}^k\}_{i=1}^n$ and $\hat{f}_{:}^k$ to this component and obtain the clip-level ($\{\vartheta_i^k\}_{i=1}^n$) features and the global context for the video ($g_\nu$).

## 2.3   Inter-Level Cooperation

By modeling the interactions between local and global context, the network learns to highlight semantics relevant to the general context of the video and to suppress the irrelevant ones: interactions between clip embeddings and the general context of the video; interactions between sentence embeddings and the general context of the text. As shown in Figure 2-Left, without knowing the global context, just from observing the frame in the third clip, there is no information about what type of *"dough"* is involved. Also the *"same woman"* in the second clip could not be related to the woman seen in the first clip.

Thus, we propose a Contextual Transformer (CoT) in Figure 2-Right to model the interactions between low-level and high-level semantics. More formally, we build the Contextual Transformer with two modules $F_{Local}$ and $F_{Global}$. We append the positional embedding to the inputs of $F_{Local}$. The goal of $F_{Local}$ is to model the short-term interactions between low-level semantics ($\{\vartheta_i^k\}_{i=1}^n$), whereas $F_{Global}$ models the interactions between local and global context ($g_\nu$) to highlight the important semantics.

Given local representations $\{\vartheta_i^k\}_{i=1}^n \in \mathbb{R}^{n \times d}$, where $n$ is the number of clips and $d$ indicates the feature dimension, $F_{Local}$ applies multi-head attention followed by a feed-forward layer and a normalization layer on top of both layers and produces embeddings $\{h_i\}_{i=1}^n$.

We compute **key (K)-value(V)** pairs based on these embeddings $\{h_i\}_{i=1}^n \in \mathbb{R}^{n \times d}$ and **query(Q)** based on the global context $g_v$. $F_{Global}$ produces the attention output as follows,

$$H_{attn} = \text{softmax}(\frac{\mathbf{QK}^T}{\sqrt{d}})\mathbf{V}, \qquad Q = \mathcal{W}_q g_v, \; K = \mathcal{W}_k \{h_i\}_{i=1}^n, \; V = \mathcal{W}_v \{h_i\}_{i=1}^n \tag{5}$$

where $\mathcal{W}_q$, $\mathcal{W}_k$, and $\mathcal{W}_v$ are the embedding weights. $H_{attn}$ is a weighted sum of values (local semantics), where the weight of each value is calculated based on its interaction with the global context query $\mathbf{Q}$. $H_{attn}$ is further encoded by a feed-forward layer to produce the contextual embedding $H_{context}$. We calculate the mean of $\{h_i\}_{i=1}^n$ and concatenate it with $H_{context}$ to obtain the final video embedding $\vartheta^k = \text{concat}(mean(\{h_i\}_{i=1}^n), H_{context})$; see Figure 2.

## 3  Cross-Modal Cycle Consistency

We introduce a cross-modal cycle-consistency loss to enforce the semantic alignment between clips and sentences, as illustrated in Figure 3. It replaces the cross-modal attention units used in [14, 8]. A pair of clip and sentence will be identified as semantically aligned if they are nearest neighbors in the learned common spaces. Consider as input a sequence of clip embeddings $\{\vartheta_i\}_{i=1}^n = \{\vartheta_1, \ldots, \vartheta_n\}$ and sentence embeddings $\{\delta_i\}_{i=1}^m = \{\delta_1, \ldots, \delta_m\}$. As the sentences of a paragraph have a temporal order, given a sentence embedding $\delta_i$ on this sequence, we first find its soft nearest neighbor [36, 37, 38]

$$\bar{\vartheta}_{\delta_i} = \sum_{j=1}^n \alpha_j \vartheta_j \quad \text{where} \quad \alpha_j = \frac{\exp(-\|\delta_i - \vartheta_j\|^2)}{\sum_{k=1}^n \exp(-\|\delta_i - \vartheta_k\|^2)} \tag{6}$$

in the clip sequence $\{\vartheta_i\}_{i=1}^n$. $\alpha_j$ is the similarity score of clip $\vartheta_j$ to sentence $\delta_i$. We then cycle back from $\bar{\vartheta}_{\delta_i}$ to the sentence sequence $\{\delta_i\}_{i=1}^m$ and calculate the soft location

$$\mu = \sum_{j=1}^m \beta_j j \quad \text{where} \quad \beta_j = \frac{\exp(-\|\bar{\vartheta} - \delta_j\|^2)}{\sum_{k=1}^m \exp(-\|\bar{\vartheta} - \delta_k\|^2)}. \tag{7}$$

The sentence embedding $\delta_i$ is semantically cycle consistent if and only if it cycles back to the original location, i.e., $i = \mu$. We penalize deviations from cycle-consistency for sampled sets of clips and sentences, which encourages the model to learn semantically consistent representations.

Our objective is the distance between the source location $i$ and the soft destination location $\mu$.

$$\ell_{CMC} = \|i - \mu\|^2 \tag{8}$$

Computing nearest neighbors as soft nearest neighbors makes the loss differentiable [36, 37, 38]. We can use this loss in both supervised and self-supervised scenarios. In the self-supervised case, we split each video uniformly into several clips and each paragraph into sentences. Beside the cycle-consistency from text to video, we also calculate $\ell_{CMC}$ from video to text. Therefore, the final $\ell_{CMC}$ loss includes both cycles.

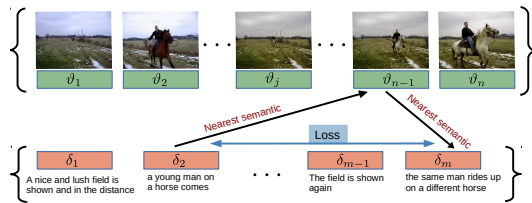

Figure 3: **Cross-Modality Cycle-Consistency.** Starting from a sentence $s_i$, we find its nearest neighbor in the clip sequence and again its neighbor in the sentence sequence. Deviations from the start index are penalized as alignment error.

The final training loss for the overall model is:

$$\ell_{final} = \ell_{align}^L + \ell_{align}^H + \ell_{align}^g + \ell_{cluster} + \lambda\ell_{CMC} \tag{9}$$

## 4  Experimental Setup

**Datasets.** We evaluate our method on the datasets **ActivityNet-captions** [39] and **Youcook2** [40]. ActivityNet-captions consists of 20k YouTube videos with an average length of 2 minutes, with 72k clip-sentence pairs. There are ~10k, ~5k and ~5k videos in train, val1 and val2, respectively. Youcook2 contains 2000 videos with a total number of 14k clips. This dataset is collected from YouTube and covers 89 types of recipes. There are ~9.6k clips for training and ~3.2k clips for validation. For each clip there is a manually annotated textual description.

**Evaluation Metrics.** We measure the performance on the retrieval task with standard retrieval metrics, i.e., recall at K (R@K e.g. R@1, R@5, R@10) and Median Rank (MR).

**Text encoding.** We feed paragraphs consisting of several sentences into a pretrained "BERT-Base, Uncased" model [19] and use the per-token outputs of the last 2 layers, resulting in 1536-d features.

**Video encoding.** For Activitynet-Captions, we use the 2048-d features provided by Zhang et al. [21] (at 3.8 FPS). For Youcook2, we test two approaches: (*A*) We follow Miech et al., 2019 [16] and concatenate 2D (Resnet-152 pretrained on ImageNet [41]) and 3D (ResNext-101 model [42] pretrained on Kinetics [43]) outputs to obtain 4096-d features at 6.2 FPS; (*B*) We use the video embedding network provided by Miech et al., 2020 [17] pretrained on video-text learning on the Howto100m dataset to obtain 512-d features at 0.9 FPS.

Table 1: **Ablation study on ActivityNet-captions (val1).** We quantify the individual contributions of the attention-aware feature aggregation (AF), the Contextual Transformer (CoT), and the cross-modal cycle-consistency loss (CMC). HSE results are reproduced by us. Disabling CoT means removing the cross-attention layer between local and global context.

| Model | Pooling Lowlvl | CMC | CoT | Paragraph $\implies$ Video | | | Video $\implies$ Paragraph | | | Param (M) |
|---|---|---|---|---|---|---|---|---|---|---|
| | | | | R@1 | R@5 | R@50 | R@1 | R@5 | R@50 | |
| HSE | Max | ✗ | ✗ | $45.6_{\pm0.3}$ | $76.1_{\pm0.7}$ | $96.0_{\pm0.3}$ | $44.9_{\pm0.5}$ | $75.8_{\pm1.2}$ | $95.8_{\pm0.4}$ | 26.1 |
| HSE | Max | ✓ | ✗ | $46.6_{\pm0.4}$ | $78.1_{\pm0.3}$ | $97.3_{\pm0.1}$ | $46.4_{\pm0.3}$ | $77.6_{\pm0.3}$ | $97.1_{\pm0.3}$ | 26.1 |
| COOT | CLS | ✗ | ✗ | $49.4_{\pm1.4}$ | $77.7_{\pm1.3}$ | $95.7_{\pm0.2}$ | $49.7_{\pm1.9}$ | $77.8_{\pm0.9}$ | $95.8_{\pm0.3}$ | 4.9 |
| COOT | AVG | ✗ | ✗ | $52.6_{\pm0.6}$ | $80.6_{\pm0.4}$ | $97.0_{\pm0.2}$ | $52.1_{\pm0.4}$ | $80.8_{\pm0.2}$ | $97.0_{\pm0.2}$ | 4.9 |
| COOT | Max | ✗ | ✗ | $58.2_{\pm0.5}$ | $84.9_{\pm0.2}$ | $98.1_{\pm0.1}$ | $58.7_{\pm0.5}$ | $86.0_{\pm0.2}$ | $98.2_{\pm0.1}$ | 4.9 |
| COOT | AFA | ✗ | ✗ | $59.0_{\pm0.5}$ | $85.4_{\pm0.2}$ | $98.2_{\pm0.0}$ | $59.8_{\pm0.6}$ | $85.8_{\pm0.8}$ | $98.2_{\pm0.1}$ | 5.8 |
| COOT | Max | ✓ | ✓ | $59.4_{\pm0.9}$ | $86.1_{\pm0.6}$ | $98.3_{\pm0.0}$ | $60.5_{\pm0.1}$ | $87.1_{\pm0.2}$ | $98.5_{\pm0.1}$ | 6.7 |
| COOT | AFA | ✗ | ✓ | $59.8_{\pm1.1}$ | $86.3_{\pm0.3}$ | $98.5_{\pm0.1}$ | $60.1_{\pm0.1}$ | $87.1_{\pm0.4}$ | $98.5_{\pm0.1}$ | 7.6 |
| COOT | AFA | ✓ | ✗ | $59.5_{\pm0.5}$ | $85.5_{\pm0.4}$ | $98.1_{\pm0.0}$ | $60.5_{\pm0.7}$ | $86.2_{\pm0.5}$ | $98.2_{\pm0.1}$ | 5.8 |
| COOT | AFA | ✓ | ✓ | $\mathbf{60.8}_{\pm0.6}$ | $\mathbf{86.6}_{\pm0.4}$ | $\mathbf{98.6}_{\pm0.1}$ | $\mathbf{60.9}_{\pm0.3}$ | $\mathbf{87.4}_{\pm0.5}$ | $\mathbf{98.6}_{\pm0.0}$ | 7.6 |

For each clip as well as for the entire video, we sample up to 80 frame features. If needed, we split the frames into 80 equal length intervals and uniformly sample a frame from each interval during training or take the center frame during validation.

**Training.** Similar to [21] we set all margins $\alpha = \alpha_g = \beta = \gamma = \mu = 0.2$. We use a mini-batch size of 64 video/paragraph pairs and sample all corresponding clips and sentences. All possible combinations of embeddings with non-matching indices in a batch are used as negative samples for the contrastive loss. To apply the cycle-consistency loss, we found that sampling 1 clip per video and 1 sentence per paragraph works best. The optimal loss weight $\lambda$ depends on architecture and dataset. As activation function, we found GELU [34] to perform best. We set the hidden size to 384 and use a pointwise linear layer to reduce the input feature dimension. We use one self-attention layer for the T-Transformer and one self-attention and one cross-attention layer for CoT. For further details on optimization and hyperparameters we refer the interested reader to the supplementary material.

**Video-Language Retrieval.** For video-text retrieval, the query is a paragraph and the task is to find the most relevant video from a database. Alternatively, the query can be a video and the task is to retrieve the most relevant paragraph. We follow the experimental protocol from Zhang et al. [21] to evaluate the models. We use the final embedding output of our model ($\vartheta^k, \delta^k$) to do the retrieval.

**Clip-sentence retrieval.** For Youcook2, we also evaluate the quality of our model when retrieving a short video clip given a single sentence. For this experiment, we use the intermediate low-level embeddings produced by our model ($\vartheta_i^k, \delta_i^k$) to do the retrieval.

## 5 Results

**Influence of each component.** We show results of a model ablation study in Table 1. First, to validate the general effectiveness of the proposed cross-modal cycle consistency loss (CMC), we apply it to the HSE architecture [21]. The $\ell_{CMC}$ loss provides a significant boost in performance for both HSE and COOT, which indicates that it will be beneficial if plugged into other video-text representation learning methods. Second, the Attention-FA module shows better performance (7.2% average improvement on R@1 for paragraph $\implies$ video and video $\implies$ paragraph tasks) than common average pooling. Third, we observe that integrating the Contextual Transformer into the overall model improves the performance. This confirms that interactions between local and global context help the model to highlight the relevant semantics (more in supp. material).

**Comparison to the state of the art** Table 2 summarizes the results of paragraph to video and video to paragraph retrieval tasks on the ActivityNet-captions dataset. For a fair comparison, our model utilizes the same video features as HSE [21]. Our method significantly outperforms all previous methods across different evaluation metrics. COOT obtains on average 16.6% better R@1 in comparison to HSE [21] while having fewer parameters. We believe the major gain comes from our attention-aware feature aggregation component and the $\ell_{CMC}$ loss.

Table 2: **Video-paragraph retrieval results on AcitvityNet-captions dataset (val1).**

| Method | Paragraph $\Longrightarrow$ Video | | | | Video $\Longrightarrow$ Paragraph | | | |
|---|---|---|---|---|---|---|---|---|
| | R@1 | R@5 | R@50 | MR | R@1 | R@5 | R@50 | MR |
| LSTM-YT [52] | 0.0 | 4.0 | 24.0 | 102.0 | 0.0 | 7.0 | 38.0 | 98.0 |
| No Context [53] | 5.0 | 14.0 | 32.0 | 78.0 | 7.0 | 18.0 | 45.0 | 56.0 |
| DENSE [39] | 14.0 | 32.0 | 65.0 | 34.0 | 18.0 | 36.0 | 74.0 | 32.0 |
| VSE [54]( [5]) | 11.7 | 34.7 | 85.7 | 10 | - | - | - | - |
| FSE [21] | 18.2 | 44.8 | 89.1 | 7 | 16.7 | 43.1 | 88.4 | 7 |
| HSE [21] | $44.4_{\pm0.5}$ | $76.7_{\pm0.3}$ | $97.1_{\pm0.1}$ | 2 | $44.2_{\pm0.6}$ | $76.7_{\pm0.3}$ | $97.0_{\pm0.3}$ | 2 |
| COOT | $\mathbf{60.8}_{\pm0.6}$ | $\mathbf{86.6}_{\pm0.4}$ | $\mathbf{98.6}_{\pm0.1}$ | **1** | $\mathbf{60.9}_{\pm0.3}$ | $\mathbf{87.4}_{\pm0.5}$ | $\mathbf{98.6}_{\pm0.0}$ | **1** |

We further provide retrieval results on the Youcook2 [40] dataset in Table 3. We compare our model under two settings: (1) with features pretrained on classification (2) with features from a pretrained SOTA video-text model.

**Without HowTo100M pretrained features.** We use features (*A*) explained in Section 4 and train the COOT model on the YouCook2 dataset. Using the same training set, COOT outperforms Miech et al. [16] and HGLMM [44] on both paragraph-to-video and sentence-to-clip tasks. This supports our rationale that modeling interactions between different hierarchy levels is crucial for capturing long-term semantics.

**With HowTo100M pretrained features.** In Table 3, we compare our method with the recently proposed SOTA methods MIL-NCE [17], ActBERT [8], and Miech et al. [16], which utilize pretraining on the huge HowTo100M dataset. We use features (*B*) (Sec. 4) and train the model on the YouCook2 dataset. Note that the paragraph to video results of other methods are computed by us. Training our model with features of a model pretrained on the HowTo100M dataset clearly improves over training with features of a model pretrained on classification and over the state-of-the-art. We can see that our model outperforms MIL-NCE [17] 16.4% on R@1 score for paragraph-to-video task, which verifies COOT benefits from hierarchy interactions. This shows that the contributions of this paper are complementary to works that focus on large-scale pretraining.

**Time complexity and number of parameters.** The COOT model has 10.6M, parameters which is 60% less than the HSE method (Table 1). Training is fast and takes less than 3 hours on two GTX1080Ti GPUs (without data I/O).

## 5.1 Video Captioning

To show that the learned representations contain meaningful information for other tasks than retrieval, we use the learned representations for video captioning building upon the captioning model MART [45]. The original method uses appearance (RGB) and optical flow features extracted from ResNet-200 [41] and BN-Inception [46], respectively.

We use the clip ($\vartheta_i^k$) and optionally the video ($\vartheta^k$) representation generated with our COOT model. In comparison to MART, we input about 100 times less features per video into the captioning model. We use the standard language evaluation metrics BLEU@3/4 [47], RougeL [48], METEOR [49], CIDEr-D [50] and R@4 [51] which measures the degree of n-gram repetition. Our results in Table 4 and Table 5 show that the MART method using our representations improves over using appearance and optical flow video features. Generated captions in Table 6 show that our video representations encapsulate richer information about the video while being more compact.

## 6 Related Work

**Image and Language**. Many self-supervised visual-language representation learning methods have focused on improving one representation with the help of the other [44, 57, 58, 59, 60, 61, 25]. These approaches learn joint image-text embeddings or map images and sentences into a common space. Recently, there has been a surging interest in utilizing Transformers for image-text representation learning [62, 63, 64, 65, 66]. ViLBERT [13] and VisualBERT [12] pretrain a BERT-like architecture on an image-text dataset and then transfer learned representations to different downstream tasks. LXMERT [14] uses additional pretraining tasks and an object-relationship component. In contrast to [14, 13], VL-BERT [15] does not utilize the task of sentence-image relationship prediction and additionally pretrains the model on text-only datasets.

Table 3: **Retrieval results on YouCook2 dataset.** Results with * are computed by us. $\triangle$ we use features of a video-text model [17] pretrained on the HowTo100m dataset.

| Method | TrainSet | Paragraph $\Longrightarrow$ Video | | | | Sentence $\Longrightarrow$ Clip | | | |
|---|---|---|---|---|---|---|---|---|---|
| | | R@1 | R@5 | R@10 | MR | R@1 | R@5 | R@10 | MR |
| Random | - | 0.21 | 1.09 | 2.19 | 229 | 0.03 | 0.15 | 0.3 | 1675 |
| Miech et al. [16] | HowTo100M | 43.1* | 68.6* | 79.1* | 2* | 6.1 | 17.3 | 24.8 | 46 |
| ActBERT [8] | HowTo100M | - | - | - | - | 9.6 | 26.7 | 38.0 | 19 |
| MIL-NCE [17] | HowTo100M | 61.9* | 89.4* | 98.9* | 1* | 15.1 | 38.0 | 51.2 | 10 |
| HGLMM [44] | YouCook2 | - | - | - | - | 4.6 | 14.3 | 21.6 | 75 |
| Miech et al. [16] | YouCook2 | 32.3* | 59.2* | 70.9* | 4* | 4.2 | 13.7 | 21.5 | 65 |
| COOT | YouCook2 | $50.4_{\pm2.6}$ | $79.4_{\pm0.6}$ | $87.4_{\pm0.8}$ | $1.3_{\pm0.6}$ | $5.9_{\pm0.7}$ | $16.7_{\pm0.6}$ | $24.8_{\pm0.8}$ | $49.7_{\pm2.9}$ |
| Miech et al. [16] | HowTo100M+ YouCook2 | 59.6* | 86.0* | 93.6* | 1* | 8.2 | 24.5 | 35.3 | 24 |
| COOT | HowTo100M$^{\triangle}$+ YouCook2 | $\mathbf{77.2}_{\pm1.0}$ | $\mathbf{95.8}_{\pm0.8}$ | $\mathbf{97.5}_{\pm0.3}$ | $\mathbf{1.0}_{\pm0.0}$ | $\mathbf{16.7}_{\pm0.4}$ | $\mathbf{40.2}_{\pm0.3}$ | $\mathbf{52.3}_{\pm0.5}$ | $\mathbf{9.0}_{\pm0.0}$ |

Table 4: **Captioning results on the YouCook2 dataset (val split).** Results with * are computed by us. $\triangle$ we use features of a video-text model [17] pretrained on the HowTo100m dataset. "MART w/o re" denotes a MART variant without recurrence.

| Features | Method | TrainSet | B@3 | B@4 | RougeL | METEOR | CIDEr-D | R@4↓ |
|---|---|---|---|---|---|---|---|---|
| RGB+Flow | VTransformer [55] | YouCook2 | 13.08* | 7.62 | 32.18* | 15.65 | 32.26 | 7.83 |
| RGB+Flow | TransformerXL [56] | YouCook2 | 11.46* | 6.56 | 30.78* | 14.76 | 26.35 | 6.30 |
| RGB+Flow | MART [45] | YouCook2 | 12.83* | 8.00 | 31.97* | 15.90 | 35.74 | **4.39** |
| COOT clip | MART | YouCook2 | 14.17 | 8.69 | 33.01 | 16.11 | 38.28 | 8.07 |
| COOT video+clip | MART | YouCook2 | 15.75 | 9.44 | 34.32 | 18.17 | 46.06 | 6.30 |
| COOT clip | MART | H100M$^{\triangle}$+YC2 | 17.12 | 10.91 | 37.59 | 18.85 | 54.07 | 5.11 |
| COOT clip | MART w/o re. | H100M$^{\triangle}$+YC2 | 17.16 | 10.69 | 37.43 | 19.18 | 54.85 | 5.45 |
| COOT clip | VTransformer | H100M$^{\triangle}$+YC2 | 17.62 | 11.09 | 37.63 | 19.34 | 54.67 | 4.57 |
| COOT video+clip | VTransformer [55] | H100M$^{\triangle}$+YC2 | 17.79 | 11.05 | 37.51 | 19.79 | 55.57 | 5.69 |
| COOT video+clip | MART | H100M$^{\triangle}$+YC2 | **17.97** | **11.30** | **37.94** | **19.85** | **57.24** | 6.69 |

**Video and Language**. The multi-modal nature of video is a great source of self-supervision. Modalities such as audio, text and motion provide strong cues to learn richer spatio-temporal features [67, 68, 16, 17, 9, 69, 11]. Aytar et al. [70] leverage natural synchronization to learn rich representations across vision, sound, and language. VideoBERT [11] learns joint video-text representations based on predicting whether the linguistic sentence is temporally aligned with the visual sentence. These approaches [8, 23, 11] focus on self-supervised pretraining and require a large set of paired video clips and texts to learn a good representation model [17].

There has been growing interest in temporal localization of natural language in videos [7, 71, 72, 73, 2]. Moment localization identifies a time window given a text query [7, 71, 74]. Most related to our work are methods that focus on joint video-text embeddings and perform video-text retrieval or captioning [4, 75, 76, 21, 9, 69, 52, 6, 5]. Several works tried to utilize the temporal structure of video and text for the alignment task [77, 70, 78, 4, 79]. Miech et al. [68] proposed a mixture of experts approach to learn text-video representations. Likewise, CE [20] proposes a mixture-of-experts model to aggregate information from pretrained experts (e.g. object, action, audio) with a gating mechanism. In our work, we use a similar hierarchy as CMHSE [21, 24, 22] and extend their design by proposing three new components to learn the interactions between different levels of the hierarchy.

**Cycle-Consistency** uses transitivity as an objective for training [38, 80, 81, 82]. The assumption of cyclical structure has been used in various works [83, 84, 85]. Wang et al. [80] obtain the supervision for visual correspondence by tracking forward and backward. Shah et al. [81] enforce consistency between the generated and the original question in visual question answering. To prevent mode collapse, cycle-consistency is activated only after a certain number of training iterations [81]. TCC [38] employs cycle-consistency for temporal video alignment. In contrast to TCC, which works only in the video domain, we align video and text. To the best of our knowledge, this is the first work which introduces cycle-consistency to the video-text domain.

Table 5: **Captioning results on the ActivityNet-Captions dataset (ae-test split of MART [45]).** Results with * are computed by us. "MART w/o re" denotes a MART variant without recurrence.

| Features | Method | TrainSet | B@3 | B@4 | RougeL | METEOR | CIDEr-D | R@4↓ |
|---|---|---|---|---|---|---|---|---|
| RGB+Flow | VTransformer [55] | ActivityNet | 16.27* | 9.31 | 29.18* | 15.54 | 21.33 | 7.45 |
| RGB+Flow | TransformerXL [56] | ActivityNet | 16.71* | 10.25 | 30.53* | 14.91 | 21.71 | 8.79 |
| RGB+Flow | MART | ActivityNet | 16.43* | 9.78 | 30.63* | 15.57 | 22.16 | 5.44 |
| COOT video+clip | TransformerXL [56] | ActivityNet | 16.94 | 10.57 | 30.93 | 14.76 | 22.04 | 15.85 |
| COOT video+clip | VTransformer [55] | ActivityNet | 16.80 | 10.47 | 30.37 | 15.76 | 25.90 | 19.14 |
| COOT clip | MART w/o re. | ActivityNet | 15.41 | 9.37 | 28.66 | 15.61 | 22.05 | 12.03 |
| COOT video+clip | MART w/o re. | ActivityNet | 16.59 | 10.33 | 29.93 | 15.64 | 25.41 | 17.03 |
| COOT clip | MART | ActivityNet | 16.53 | 10.22 | 30.68 | 15.91 | 23.98 | **5.35** |
| COOT video+clip | MART | ActivityNet | **17.43** | **10.85** | **31.45** | **15.99** | **28.19** | 6.64 |

Table 6: **Captioning samples, more accurate (left) and less accurate (right) cases.** First row: ActivityNet (ae-test split), second row: YouCook2 (val split). Red/bold indicates content errors, blue/italic indicates repetitive patterns.

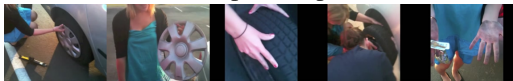
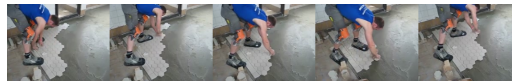

**MART**: A person is **driving the car**. **A boy** is **holding a bottle of wood**.

**COOT (Ours)**: A woman is seen kneeling down next to a car while others stand around her. The woman then pushes the tire **back and fourth**.

**GT**: A girl is shown trying to change a tire. She successfully removes the tire, then replaces it with a spare, showing off their dirty hands afterward.

**MART**: A man *is kneeling down on* a floor. He *is kneeling down on* the ground.

**COOT (Ours)**: A man is seen kneeling down on the ground and begins **putting shoes on**. The man continues to put on the shoes and ends by *putting his shoes on*.

**GT**: A person is seen bending over a floor placing tiles down over the plaster. The person continues laying tiles down and pushing down on the floor to make sure it's sturdy.

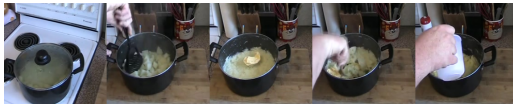
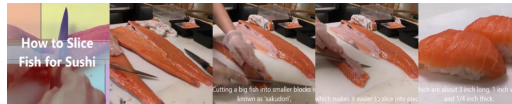

**MART**: Heat up a pan and cook until golden brown. Add **onions** *to the pan*. Add flour salt and pepper *to the pan*. Add rice *to the pan* and stir.

**COOT (Ours)**: Boil the potatoes in water. Add chopped *potatoes* to the pan. Add butter and mash. Add some milk and *mash*.

**GT**: Boil some small pieces of potatoes in water. Mash the potato. Add some butter and salt and stir. Gradually add milk while stirring the potatoes.

**MART**: Cut the salmon into thin slices. *Cut the salmon into thin slices. Cut the salmon into thin slices. Cut the salmon into thin slices.*

**COOT (Ours)**: Cut the salmon in half. *Cut the salmon in half. Cut the salmon* into thin slices. *Cut the salmon into thin slices.*

**GT**: Slice the fish into smaller pieces. Chop the tail end off. Cut the fish at an angle. Cut the fish into thin pieces.

## 7 Conclusions

We have presented a cooperative hierarchical transformer architecture for learning a joint video and text embedding space where similar semantics are aligned. The architecture is designed to encourage the use of long-range temporal context in a cross-level manner. Our approach uses two new components to model the interactions within and between hierarchy levels; an attention-aware feature aggregation module to model the interactions between frames and words, a contextual transformer to model the interactions between local contexts and global context. In addition, we have introduced a new cross-modal cycle-consistency loss which enforces the semantic alignment of clips and sentences. We have shown that both components contribute – jointly and individually – to an improved retrieval performance. As a result, our approach achieves state-of-the-art retrieval and captioning performance on two challenging datasets.

## Broader Impact

This work contributes fundamental research and does not present any foreseeable societal consequence. In the long run, this line of research can contribute to services on video search and video organisation.

## Acknowledgments

We thank Ehsan Adeli for helpful comments, Antoine Miech for providing details on their retrieval evaluation, and Facebook for providing us a GPU server with Tesla P100 processors for this research work.

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
