[Supplementary Material · coot_supp_final.pdf]

# COOT: Cooperative Hierarchical Transformer for Video-Text Representation Learning –Supplementary Material–

**Simon Ging**[1][*], **Mohammadreza Zolfaghari**[1][*], **Hamed Pirsiavash**[2], **Thomas Brox**[1]
[1]University of Freiburg, [2]University of Maryland Baltimore County
[1]{gings, zolfagha, brox}@cs.uni-freiburg.de, [2] hpirsiav@umbc.edu

## 1  Implementation Details

**Hyperparameters.**   To select hyperparameters for our model, we used a combination of manual search and BOHB [1] to explore the hyperparameters space. In Table 1, we provide an overview of our hyperparameter search.

After testing different activation functions ReLU, SELU [2], ELU [3] and GELU [4] we found GELU to perform best. Increasing the capacity of the model by using a higher attention dimension can help improve the results, but comes at the cost of higher memory requirements and more difficult optimization.

Table 1: **Hyperparameters.** This table shows the hyperparameter ranges we considered and the final choices for our three best models (ActivityNet-captions, YouCook2-Resnet/Resnext features, Youcook2-Howto100m features.). ROP denotes the *Reduce on Plateau* Scheduler we used. Dimensions given in multiples (1x, 2x) refer to multiples of the Attention Dimension parameter. FF denotes Feed-Forward. AF is our Attention-aware Feature Aggregation module.

| Hyperparameter | Considered Range | | ActivityNet | Youcook2 | |
|---|---|---|---|---|---|
| | | | | Resnet/ResneXt | Howto100m |
| Optimizer | Adam, RAdam, SGD | | Adam | RAdam | RAdam |
| Learning rate | 1e-5 | 1e-2 | 1e-3 | 3.6e-4 | 9e-4 |
| Weight Decay | 0 | 1e-2 | 2e-5 | 2e-5 | 0 |
| Momentum | 0.5 | 0.99 | 0.9 | 0.56 | 0.56 |
| Adam Beta2 | 0.9 | 0.9999 | 0.999 | 0.98 | 0.98 |
| Adam Epsilon | 1e-10 | 1e-7 | 1e-8 | 1.5e-9 | 1.5e-9 |
| Warmup Epochs | 0 | 8 | 3 | 0 | 0 |
| ROP Patience | 2 | 10 | 2 | 5 | 5 |
| ROP Cooldown | 0 | 3 | 3 | 3 | 3 |
| Attention Layers | 1 | 3 | 1 | 1 | 1 |
| Attention Dimension | 256 | 1024 | 384 | 384 | 384 |
| Attention Heads | 1 | 8 | 8 | 8 | 8 |
| Attention FF Dimension | 1x | 2x | 1x | 1x | 1x |
| AF Dimension | 1x | 2x | 2x | 2x | 2x |
| AF Heads | 1 | 8 | 2 | 2 | 2 |
| Number of AF modules | 1 | 2 | 1 | 1 | 1 |
| Dropout | 0% | 10% | 2.5% | 1% | 5% |
| Gaussian Noise on Frame Features | 0 | 1 | 0 | 0.01 | 0 |

**Optimization.**   We tried several optimizers such as Adam, RAdam [5] and SGD. If carefully configured, RAdam can improve over Adam.

We schedule the Learning Rate with a *Reduce on Plateau* approach: Whenever our validation metric does not improve for a certain number of epochs, we reduce the learning rate by a factor of

Table 2: **Text feature ablation study on ActivityNet-captions (val1).** We evaluate our choice of text encoding and show that Bert [6] outperforms GloVe [7] on both models and all metrics.

| Model | Text | Paragraph $\Longrightarrow$ Video | | | Video $\Longrightarrow$ Paragraph | | |
|---|---|---|---|---|---|---|---|
| | | R@1 | R@5 | R@50 | R@1 | R@5 | R@50 |
| HSE | GloVe | $45.7_{\pm 0.3}$ | $76.1_{\pm 0.7}$ | $96.0_{\pm 0.3}$ | $44.9_{\pm 0.5}$ | $75.8_{\pm 1.2}$ | $95.8_{\pm 0.4}$ |
| HSE | Bert | $47.0_{\pm 1.1}$ | $77.0_{\pm 1.5}$ | $96.1_{\pm 0.4}$ | $46.9_{\pm 0.8}$ | $77.2_{\pm 1.1}$ | $95.9_{\pm 0.6}$ |
| COOT | GloVe | $56.5_{\pm 1.1}$ | $84.1_{\pm 1.3}$ | $98.0_{\pm 0.3}$ | $57.3_{\pm 1.8}$ | $84.5_{\pm 1.4}$ | $98.2_{\pm 0.2}$ |
| COOT | Bert | $\mathbf{60.8}_{\pm 0.6}$ | $\mathbf{86.6}_{\pm 0.4}$ | $\mathbf{98.6}_{\pm 0.1}$ | $\mathbf{60.9}_{\pm 0.3}$ | $\mathbf{87.4}_{\pm 0.5}$ | $\mathbf{98.6}_{\pm 0.0}$ |

10. After no improvements for 15 epochs, we terminate the training process. As relevant metric we defined the sum of R@1 Retrieval Score for video-paragraph and paragraph-video retrieval on Activitynet-Captions and the sum of R@1 Retrieval Score for clip-sentence and sentence-clip retrieval on Youcook2. Careful tuning of the optimizer parameters, using an automated search method like BOHB [1] to search parts of the parameters space, was crucial to train the models properly.

**Value of $\lambda$:** For Activitynet we used 0.01 and for Youcook2 0.001.

For weight initialization, we utilized Uniform, Normal and Truncated Normal distributions. The best results were obtained with initializing weights randomly from the Truncated Normal distribution with a standard deviation of 0.01, redrawing all samples with more than 2 standard deviations.

To cope with the overfitting problem, the different regularization methods (Dropout, Weight Decay, CMC-loss, Gaussian Noise on Frame Features) need to be traded off carefully to obtain good results (see Table 1).

**Preprocessing.** For ActivityNet captions, we found it helpful to expand all clips to be at least 10 frames long. Expanding is done by iteratively adding frames to the start and end of the clip until we reach the desired length.

**Retrieval.** We L2-normalize the output embeddings of our model so the squared elements sum to 1. Retrieval is done by cosine similarity, e.g. given video embedding $v$, we retrieve paragraph embedding

$$p = \max_{\hat{p} \in \mathcal{D}} v^\top \hat{p} \tag{1}$$

## 2 Experiments

In this section, we provide ablation studies on the importance of low-level supervision, different text encoders performance, impact of different alignment losses in our final training loss and analysis on sequence pooling.

### 2.1 Ablation Study

**Importance of low-level supervision.** In Fig. 1, we study the effect of adding uniform noise to the start and end frame index of each clip in ActivityNet-captions from the interval $[-N_f * P, +N_f * P]$. $N_f$ is the total number of video frames and $P$ is the noise percentage. We also perform a "full" noise experiment where we drop the temporal alignment labels of clips and sentences completely. We observe that increasing the noise from 0% to 40% consistently decreases the performance as labels get less reliable. For noise more than 40%, we do not observe significant changes in performance anymore. This is probably because at this noise level the labels become useless and are ignored. Still a good performance is obtained.

The study shows that the model is robust to noisy and missing low-level supervision and COOT is still able to capture useful dynamics between low-level and high-level semantics.

**Impact of Text Encoding.** We conduct ablation experiments to evaluate the importance of the text encoder for representation learning task. The ablation study results are shown in Table 2. We

Table 3: **Loss function ablation study on ActivityNet-captions (val1).** We analyse performance of the COOT model while removing loss components with different base models. CoT denotes using global attention in the contextual transformer. AF is our Attention-aware Feature Aggregation module.

| # | Pooling Lowlvl | CMC | CoT | Alignment | | | Clustering | | Par. $\Longrightarrow$ Video | | Video $\Longrightarrow$ Par. | |
|---|---|---|---|---|---|---|---|---|---|---|---|---|
| | | | | High | Low | Ctx | High | Low | R@1 | R@5 | R@1 | R@5 |
| 1 | Avg | ✗ | ✗ | ✓ | ✗ | ✗ | ✗ | ✗ | $30.4_{\pm3.2}$ | $58.4_{\pm4.5}$ | $29.9_{\pm3.3}$ | $58.7_{\pm4.5}$ |
| 2 | Avg | ✗ | ✗ | ✓ | ✗ | ✓ | ✓ | ✗ | $49.7_{\pm0.7}$ | $79.0_{\pm0.6}$ | $48.6_{\pm0.5}$ | $79.1_{\pm0.9}$ |
| 3 | Avg | ✗ | ✗ | ✓ | ✗ | ✓ | ✓ | ✓ | $49.2_{\pm0.7}$ | $78.9_{\pm0.2}$ | $48.6_{\pm0.6}$ | $78.9_{\pm0.6}$ |
| 4 | Avg | ✗ | ✗ | ✓ | ✓ | ✗ | ✓ | ✓ | $50.6_{\pm1.1}$ | $79.8_{\pm0.8}$ | $50.8_{\pm1.0}$ | $79.8_{\pm0.8}$ |
| 5 | Avg | ✗ | ✗ | ✓ | ✓ | ✓ | ✗ | ✗ | $51.5_{\pm0.7}$ | $80.2_{\pm0.4}$ | $52.0_{\pm0.8}$ | $80.5_{\pm0.3}$ |
| 6 | Avg | ✗ | ✗ | ✓ | ✓ | ✓ | ✓ | ✓ | $52.6_{\pm0.6}$ | $80.6_{\pm0.4}$ | $52.1_{\pm0.4}$ | $80.8_{\pm0.2}$ |
| 7 | Avg | ✓ | ✗ | ✓ | ✗ | ✗ | ✗ | ✗ | $27.4_{\pm2.1}$ | $55.3_{\pm2.4}$ | $27.3_{\pm1.6}$ | $56.0_{\pm2.3}$ |
| 8 | Avg | ✓ | ✗ | ✓ | ✓ | ✓ | ✓ | ✓ | $54.1_{\pm0.8}$ | $82.0_{\pm0.1}$ | $54.7_{\pm0.2}$ | $82.1_{\pm0.1}$ |
| 9 | Avg | ✓ | ✓ | ✓ | ✓ | ✓ | ✓ | ✓ | $53.6_{\pm0.1}$ | $81.7_{\pm0.0}$ | $53.5_{\pm0.5}$ | $81.7_{\pm0.7}$ |
| 10 | Max | ✗ | ✗ | ✓ | ✗ | ✗ | ✗ | ✗ | $47.9_{\pm0.7}$ | $76.9_{\pm0.1}$ | $48.3_{\pm0.2}$ | $77.5_{\pm0.6}$ |
| 11 | Max | ✗ | ✗ | ✓ | ✗ | ✓ | ✓ | ✗ | $56.5_{\pm0.3}$ | $84.5_{\pm0.2}$ | $56.6_{\pm0.4}$ | $85.2_{\pm0.1}$ |
| 12 | Max | ✗ | ✗ | ✓ | ✗ | ✓ | ✓ | ✓ | $54.4_{\pm0.9}$ | $83.3_{\pm0.9}$ | $55.4_{\pm1.4}$ | $84.0_{\pm0.8}$ |
| 13 | Max | ✗ | ✗ | ✓ | ✓ | ✗ | ✓ | ✓ | $55.3_{\pm0.8}$ | $83.0_{\pm0.8}$ | $56.4_{\pm1.1}$ | $83.7_{\pm1.2}$ |
| 14 | Max | ✗ | ✗ | ✓ | ✓ | ✓ | ✗ | ✗ | $56.1_{\pm0.2}$ | $83.3_{\pm0.2}$ | $57.0_{\pm0.3}$ | $83.9_{\pm0.5}$ |
| 15 | Max | ✗ | ✗ | ✓ | ✓ | ✓ | ✓ | ✓ | $58.2_{\pm0.5}$ | $84.9_{\pm0.2}$ | $58.7_{\pm0.5}$ | $86.0_{\pm0.2}$ |
| 16 | Max | ✓ | ✗ | ✓ | ✗ | ✗ | ✗ | ✗ | $46.3_{\pm1.0}$ | $76.2_{\pm0.9}$ | $47.7_{\pm0.9}$ | $77.2_{\pm0.7}$ |
| 17 | Max | ✓ | ✗ | ✓ | ✓ | ✓ | ✓ | ✓ | $57.5_{\pm0.5}$ | $84.8_{\pm0.2}$ | $58.1_{\pm1.0}$ | $85.3_{\pm0.4}$ |
| 18 | Max | ✓ | ✓ | ✓ | ✓ | ✓ | ✓ | ✓ | $59.4_{\pm0.9}$ | $86.1_{\pm0.6}$ | $60.5_{\pm1.0}$ | $87.1_{\pm0.2}$ |
| 19 | AF | ✗ | ✗ | ✓ | ✗ | ✗ | ✗ | ✗ | $47.1_{\pm0.7}$ | $76.7_{\pm0.6}$ | $47.6_{\pm0.2}$ | $77.4_{\pm0.2}$ |
| 20 | AF | ✗ | ✗ | ✓ | ✗ | ✓ | ✓ | ✗ | $56.3_{\pm0.3}$ | $84.0_{\pm0.2}$ | $56.8_{\pm0.7}$ | $84.7_{\pm0.3}$ |
| 21 | AF | ✗ | ✗ | ✓ | ✗ | ✓ | ✓ | ✓ | $55.2_{\pm0.2}$ | $83.3_{\pm0.1}$ | $55.8_{\pm0.5}$ | $83.6_{\pm0.2}$ |
| 22 | AF | ✗ | ✗ | ✓ | ✓ | ✗ | ✓ | ✓ | $57.8_{\pm0.3}$ | $84.6_{\pm0.2}$ | $58.1_{\pm0.3}$ | $85.1_{\pm0.2}$ |
| 23 | AF | ✗ | ✗ | ✓ | ✓ | ✓ | ✗ | ✗ | $58.8_{\pm0.4}$ | $85.3_{\pm0.4}$ | $59.1_{\pm0.6}$ | $85.8_{\pm0.4}$ |
| 24 | AF | ✗ | ✗ | ✓ | ✓ | ✓ | ✓ | ✓ | $59.0_{\pm0.5}$ | $85.4_{\pm0.2}$ | $59.8_{\pm0.6}$ | $85.8_{\pm0.8}$ |
| 25 | AF | ✓ | ✗ | ✓ | ✗ | ✗ | ✗ | ✗ | $47.8_{\pm0.5}$ | $76.4_{\pm0.4}$ | $47.8_{\pm0.2}$ | $77.5_{\pm0.5}$ |
| 26 | AF | ✓ | ✗ | ✓ | ✓ | ✓ | ✓ | ✓ | $59.5_{\pm0.5}$ | $85.5_{\pm0.4}$ | $60.5_{\pm0.7}$ | $86.2_{\pm0.5}$ |
| 27 | AF | ✓ | ✓ | ✓ | ✗ | ✗ | ✗ | ✗ | $53.9_{\pm0.7}$ | $82.6_{\pm0.6}$ | $53.8_{\pm0.6}$ | $83.0_{\pm0.5}$ |
| 28 | AF | ✓ | ✓ | ✓ | ✓ | ✓ | ✗ | ✗ | $55.1_{\pm5.3}$ | $83.4_{\pm3.6}$ | $55.5_{\pm4.7}$ | $83.8_{\pm3.2}$ |
| 29 | AF | ✓ | ✓ | ✓ | ✓ | ✗ | ✓ | ✓ | $58.5_{\pm1.1}$ | $85.2_{\pm0.5}$ | $58.5_{\pm0.7}$ | $85.5_{\pm0.7}$ |
| 30 | AF | ✓ | ✓ | ✓ | ✓ | ✓ | ✓ | ✓ | $\mathbf{60.8}_{\pm0.6}$ | $\mathbf{86.6}_{\pm0.4}$ | $\mathbf{60.9}_{\pm0.3}$ | $\mathbf{87.4}_{\pm0.5}$ |

first evaluate the COOT model and the HSE [8] model with GloVe [7] features. We then replace GloVe features with features obtained from a pretrained Bert [6] model. Note that we feed an entire paragraph consisting of several sentences into Bert, leveraging high-level context. Our results show that replacing fixed word embeddings with the context-aware Bert features can significantly improve model performance over different architectures. Both models are relatively shallow (1 layer of attention / GRU respectively), which may be the reason why the deeper Bert model (13 layers) can help understand the text better.

**Impact of Alignment Losses.** We study the effect of alignment losses on the performance in Table 3. To give a more diverse picture, we evaluate the losses under different settings: We use three different low level pooling methods (Averagepool, Maxpool and Attention-aware Feature Aggregation) and selective disable the Cross-Modal Cycle Consistency loss and the global context attention in the Contextual Transformer.

We found that removing any or all of the three alignment losses significantly decreases performance. In addition, we observed that clustering losses have a positive impact on the performance of the model.

Note that we also tried clustering the global context and found it to be not helpful. It might be a too strong constraint on our low-level embedding network.

**Study on sequence pooling methods.** In Table 4 we replace our low-level (frames, words) and high-level (clips, sentences) pooling methods and evaluate the performance. Interestingly, we get

Table 4: **Evaluation of different sequence pooling methods on ActivityNet-captions (val1).** We switch both the low level (frames, words) and high level (clips, sentences) pooling methods and observe the changes in performance. In experiments denoted with *, we used a different optimizer setting.

| Pooling Low | High | CMC | CoT | Par. ⟹ Video R@1 | R@5 | Video ⟹ Par. R@1 | R@5 |
|---|---|---|---|---|---|---|---|
| AF | AF x1 | ✓ | ✓ | $42.6_{\pm0.4}$ | $76.5_{\pm0.3}$ | $42.1_{\pm0.8}$ | $76.9_{\pm0.7}$ |
| AF | AF x2 | ✓ | ✓ | $42.8_{\pm0.1}$ | $76.0_{\pm0.7}$ | $42.8_{\pm0.1}$ | $76.4_{\pm0.6}$ |
| AF* | AF x1 | ✓ | ✓ | $48.7_{\pm1.0}$ | $82.2_{\pm0.6}$ | $50.1_{\pm0.4}$ | $82.6_{\pm0.4}$ |
| AF* | AF x2 | ✓ | ✓ | $50.5_{\pm0.4}$ | $82.3_{\pm0.4}$ | $51.4_{\pm1.4}$ | $82.9_{\pm0.6}$ |
| Max | Max | ✓ | ✓ | $40.9_{\pm0.7}$ | $75.3_{\pm0.1}$ | $42.2_{\pm0.5}$ | $76.2_{\pm0.6}$ |
| AF | Max | ✓ | ✓ | $43.3_{\pm0.9}$ | $76.3_{\pm1.0}$ | $42.5_{\pm0.6}$ | $77.2_{\pm1.2}$ |
| CLS | Avg | ✗ | ✗ | $49.4_{\pm1.4}$ | $77.7_{\pm1.3}$ | $49.7_{\pm1.9}$ | $77.8_{\pm0.9}$ |
| CLS | Avg | ✓ | ✓ | $49.7_{\pm0.5}$ | $79.4_{\pm0.2}$ | $51.2_{\pm0.1}$ | $79.6_{\pm0.1}$ |
| Avg | Avg | ✗ | ✗ | $52.6_{\pm0.6}$ | $80.6_{\pm0.4}$ | $52.1_{\pm0.4}$ | $80.8_{\pm0.2}$ |
| Avg | Avg | ✓ | ✓ | $53.6_{\pm0.1}$ | $81.7_{\pm0.0}$ | $53.5_{\pm0.5}$ | $81.7_{\pm0.7}$ |
| Max | Avg | ✗ | ✗ | $58.2_{\pm0.5}$ | $84.9_{\pm0.2}$ | $58.7_{\pm0.5}$ | $86.0_{\pm0.2}$ |
| Max | Avg | ✓ | ✓ | $59.4_{\pm0.9}$ | $86.1_{\pm0.6}$ | $60.5_{\pm1.0}$ | $87.1_{\pm0.2}$ |
| AF | Avg | ✓ | ✓ | $\mathbf{60.8}_{\pm0.6}$ | $\mathbf{86.6}_{\pm0.4}$ | $\mathbf{60.9}_{\pm0.3}$ | $\mathbf{87.4}_{\pm0.5}$ |

Table 5: **Evaluation of different averagepooling methods.** We modify our exact approach to averagepooling in the high-level and evaluate the results.

**ActivityNet-captions** dataset:

| Sum | Pad | Divide | Par. ⟹ Video R@1 | R@5 | Video ⟹ Par. R@1 | R@5 |
|---|---|---|---|---|---|---|
| All | Max(Batch, 16) | All | $44.2_{\pm2.5}$ | $75.4_{\pm2.5}$ | $44.1_{\pm2.0}$ | $75.9_{\pm2.0}$ |
| All | Max(Batch, 16) | Nonzero | $44.1_{\pm0.7}$ | $76.2_{\pm0.9}$ | $44.7_{\pm1.1}$ | $76.9_{\pm0.9}$ |
| All | Batch | All | $48.3_{\pm0.2}$ | $76.8_{\pm0.8}$ | $47.9_{\pm0.8}$ | $77.7_{\pm0.7}$ |
| Nonzero | Batch | Nonzero | $42.0_{\pm0.5}$ | $76.3_{\pm0.3}$ | $41.6_{\pm0.6}$ | $77.0_{\pm0.7}$ |
| All | Batch | Nonzero | $\mathbf{60.8}_{\pm0.6}$ | $\mathbf{86.6}_{\pm0.4}$ | $\mathbf{60.9}_{\pm0.3}$ | $\mathbf{87.4}_{\pm0.5}$ |

**Youcook2** dataset:

| Sum | Pad | Divide | Par. ⟹ Video R@1 | R@5 | Sent. ⟹ Clip R@1 | R@5 |
|---|---|---|---|---|---|---|
| All | Max(Batch, 16) | All | $\mathbf{77.6}_{\pm0.7}$ | $\mathbf{96.3}_{\pm0.4}$ | $\mathbf{17.5}_{\pm0.3}$ | $\mathbf{40.7}_{\pm0.1}$ |
| All | Max(Batch, 16) | Nonzero | $74.7_{\pm2.0}$ | $95.0_{\pm0.6}$ | $16.9_{\pm0.5}$ | $39.7_{\pm0.8}$ |
| All | Batch | All | $77.4_{\pm1.5}$ | $96.2_{\pm1.6}$ | $17.2_{\pm0.6}$ | $39.9_{\pm0.3}$ |
| Nonzero | Batch | Nonzero | $74.2_{\pm2.6}$ | $94.7_{\pm0.7}$ | $16.8_{\pm0.3}$ | $40.2_{\pm0.6}$ |
| All | Batch | Nonzero | $77.2_{\pm1.0}$ | $95.8_{\pm0.8}$ | $16.7_{\pm0.4}$ | $40.2_{\pm0.3}$ |

the best results with our AF module on the low level, while averagepooling outperforms it on the high level. Removing our components (CMC, CoT) and replacing AF with maxpooling provides a considerably strong baseline compared to our full model.

The sequence length is higher on the low level (e.g. up to 80 frames) than on the high level (on average 3.6 clips per video). Additionally, there are stronger temporal relationships between semantics in the low level. The AF module can learn to capture these relationships and improve the features. However on the high-level, the semantics have more independent meanings which makes it much harder for AF to model the temporal relationships between them.

Note that to give a more fair comparison, we change the optimizer setting when adding AF on the high level, as denoted with *. We observe that concatenating the output of 2 AF modules on the high level improves the performance, suggesting that the two modules learn to attend to different features.

Table 6: **Video-paragraph retrieval results on AcitvityNet-captions dataset (val2).**

| Method | Par. $\Longrightarrow$ Video | | Video $\Longrightarrow$ Par. | |
|---|---|---|---|---|
| | R@1 | R@5 | R@1 | R@5 |
| FSE | 11.5 | 31.0 | 11.0 | 30.6 |
| HSE | 32.9 | 62.7 | 32.6 | 63.0 |
| COOT | **48.5** | **78.9** | **48.9** | **79.5** |

We also vary our approach on averagepooling on the high level and report results In Table 5. Working on variable length inputs, there are a number of design choices to make. We evaluate the following ones: **A)** Summing over the unmasked sequence elements (nonzero inputs) only or summing over both sequence and padding elements (zero inputs). **B)** Minimum padding to the maximum sequence length in the minibatch or to a length of at least 16. **C)** Obtaining the average by dividing the sum by the length of nonzero elements or by the length of all elements.

On **ActivityNet-captions** (split val1, average sequence length 3.6), we show our non-standard approach of including padding tokens in the sum but dividing by the length of non-padding tokens works well. Note that in all other reported experiments, we use this version of averagepooling.

On **Youcook2** (split val, average sequence length 7.6), we cannot reproduce this large gain in performance but the approach still works reasonably well. The good results when padding to a minimum length of 16 might be due to the average length being closer to 16 than in ActivityNet-captions.

Figure 1: Noise vs Performance study on ActivityNet-captions dataset (val1)

## 2.2 Retrieval on ActivityNet-captions (split val2)

We provide retrieval results for ActivityNet-Captions (val2 split) in Table 6.

# 3 Qualitative Results

**ActivityNet-Captions.** To further check whether our COOT model can learn the semantic alignment of video and text well, we provide qualitative examples for the retrieval task on the ActivityNet-caption dataset (val1 split, 4917 video-paragraph pairs). Note that any spelling errors in the dataset are not corrected. As shown in Table 7 and Table 8, the model learns to semantically align the video and paragraph embeddings. Even for imperfect rankings, the model retrieves semantically similar items.

**YouCook2.** We also present a set of qualitative clip-to-sentence and sentence-to-clip retrieval examples for the YouCook2 dataset (val split, 3492 clip-sentence pairs, 457 video-paragraph pairs). Table 9 and Table 10 show several examples where we can reasonably retrieve similar semantics, even when the wrong object is recognized (Table 9-Right).

**t-SNE Visualization of Embeddings.** We project the video embeddings of Activitynet dataset to 2D space using t-SNE [9] and visualize each point with a sample frame from the video. As shown in Figure 2, the embeddings are clustered semantically around activities and videos with similar content are in close neighborhood.

Table 7: **Qualitative Results on Activitynet for Paragraph-to-Video Retrieval.** For each text query, we show some frames from the top three ranked videos together with the correct video. For clarification, we show video results with text. **Left:** The correct video has a high rank and all top results are very relevant to the query. **Right:** Even though the correct video is ranked low, the top videos are semantically similar to the text query.

| *Query:* A man is standing inside a workshop. He leans over, welding a piece of metal. Sparks fly as he welds. | | *Query:* A person is kneeling down painting something on the ground. They smooth out the paint. They continue painting layers on top of the paint. | |
| --- | --- | --- | --- |
| Rank *Score* | Retrieved Video | Rank *Score* | Retrieved Video |
| **1** *0.827* |  A man approaches a table with his soldiering gun. The man soldiers on a piece of metal. The man stops and walks away from his table. | **1** *0.654* |  man is in a living room painting a couch with purle spray. man paint the cushions of the couch on top of paperboard. |
| **2** *0.821* |  A man is standing inside a workshop. He leans over, welding a piece of metal. Sparks fly as he welds. | **2** *0.643* |  A white yard chair is being shown. A man scrapes the paint off the chair with a scraper. He shows off a bucket of paint, and uses the white paint to coat the chair. |
| **3** *0.816* |  A man is welding something on a table. He is moving his hand while he's welding. He finishes welding and lifts up his mask. | **3** *0.640* |  A woman is seated on a work table and holds a paint brush. The woman paints a picture of long stems and leaves of a plant. The woman paints flower petals onto the painting. |
| **4** *0.783* |  A man in a brown shirt is standing in a room. He is wearing a mask and welding something. He stops welding and starts welding the back of it. | **48** *0.438* |  A person is kneeling down painting something on the ground. They smooth out the paint. They continue painting layers on top of the paint. |

# 4   Captioning Results

To expand upon the qualitative captioning results, we provide evaluation on samples that are not cherry-picked for Youcook2 (val split) and ActivityNet (ae-val and ae-test split) in Tables 11, 12, 13.

## Acknowledgments and Disclosure of Funding

Use unnumbered first level headings for the acknowledgments. All acknowledgments go at the end of the paper before the list of references. Moreover, you are required to declare funding (financial activities supporting the submitted work) and competing interests (related financial activities outside the submitted work). More information about this disclosure can be found at: `https://neurips.cc/Conferences/2020/PaperInformation/FundingDisclosure`.

Do **not** include this section in the anonymized submission, only in the final paper. You can use the `ack` environment provided in the style file to autmoatically hide this section in the anonymized submission.

Table 8: **Retrieval Video to Paragraph on Activitynet.** Long paragraphs have been shortened, as indicated by "[...]". **Left:** The correct paragraph is identified with a considerable score margin to the 2nd place. **Right:** The top results are from the same activity as the input video (*dancing*).

| *Query:* | | *Query:* | |
|---|---|---|---|
|  | |  | |
| **Rank** *Score* | Retrieved Text | **Rank** *Score* | Retrieved Text |
| **1** *0.813* | **A woman is resting next to crashing water. She is smoking a pipe. She blows out a plume of smoke.** | **1** *0.717* | A woman stands in front of a crowd of people on a public sidewalk and dances with a male dance partner in ballroom style dance. [...] |
| **2** *0.654* | A close up of a man's chin is shown followed by him smoking a hookah pipe. He takes the pipe out of his mouth and blows the smoke into the camera. | **2** *0.678* | A woman in a leather dress and hat dances in a public station. A man joins her, dancing side to side in a flamenco style dance. They continue dancing as a small crowd gathers to watch. [...] |
| **3** *0.641* | A close up of tin foil is shown leading a woman taking a large hit out of a hookah hose. She continues smoking out of the hookah [...] | **3** *0.608* | A large group of people are seen standing around a city center waiting for people to arrive. Girls dancing are seen walking through the parade as other people watch on the side. [...] |
| **4** *0.601* | A woman is laying back in a chair getting her lip pierced. The piercer removes the tool and pulls on her lip. | **16** *0.496* | **People are dancing in a street. People are standing on the sidelines watching them. They continue dancing on a street.** |

Table 9: **Sentence-to-Clip Retrieval on Youcook2.** For clarification, we show clip results with corresponding text. **Left:** The model ranks the correct video at the top and even distinguishes it from other videos about the same activity. **Right:** The *slicing* task is correctly recognized, but the model is not able to understand which object is being chopped (*bamboo shots*).

| *Query:* melt butter in the pan | | *Query:* slice the bamboo shoots into strips | |
|---|---|---|---|
| Rank<br>*Score* | Retrieved Clip | Rank<br>*Score* | Retrieved Clip |
| **1**<br>*0.642* | <br>melt butter in the pan | **1**<br>*0.621* | <br>finely chop a bundle of parsley and add to a bowl |
| **2**<br>*0.583* | <br>melt the butter in a pan | **2**<br>*0.610* | <br>finely chop green onions |
| **3**<br>*0.561* | <br>melt the butter in a pan | **3**<br>*0.609* | <br>cut garlic carrots celery onion and bok choi into thin slices |
| **4**<br>*0.553* | <br>heat the butter and some sea salt flakes in the pan | **168**<br>*0.326* | <br>slice the bamboo shoots into strips |

Table 10: **Clip-to-Sentence Retrieval on Youcook2 val set. Left:** The model gives high relative score to the relevant text but has problems visually distinguishing *apples* from *potatoes*. **Right:**: *Wine* is confused with *oil* and the herbs cannot be identified precisely to be *bay leaves* and *thyme*. Identical sentences can produce different results, since the Bert [6] text encoder takes paragraph context into account and therefore the model inputs differ.

| Query: | | Query: | |
|---|---|---|---|
|  | |  | |
| Rank *Score* | Retrieved Text | Rank *Score* | Retrieved Text |
| **1** 0.523 | place the potato wedges into a pan of hot oil | **1** 0.705 | add oil and herbs to a pan |
| **2** 0.514 | **cook the apple slices in the pan** | **2** 0.622 | heat oil to 365 in a pan |
| **3** 0.510 | remove the potatoes from the oil and place on paper towel | **3** 0.603 | heat some oil in a pan |
| **4** 0.497 | add oil to the pan and fry the hash browns | **4** 0.579 | heat some oil in a pan |
| **5** 0.495 | fry the potatos in oil | **5** 0.575 | add oil to a pan |
| **6** 0.480 | add the potatoes to the pan | **6** 0.570 | heat some olive oil in a pan |
| **7** 0.477 | heat the apple in a pan with some oil | **7** 0.567 | heat some oil in a pan |
| **8** 0.475 | pierce the knife inside the potatoes and find if the potatoes are cooked properly | **8** 0.564 | heat oil in a pan |
| **9** 0.474 | melt little butter and olive oil in a pan | **9** 0.564 | heat some oil cumin seeds and coriander seeds in a pan |
| **10** 0.470 | fry the potatoes in a deep fryer | **85** 0.385 | **add white wine onions a bay leaf and thyme to the pot** |

Table 11: **Random Captioning samples on YouCook2 (val split).**

**MART**: Cook the bacon in a pan. Add chopped onions to the pan. Add chopped carrots. Add chopped tomatoes to the pan. Add the chicken to the pan.

**COOT (Ours)**: Fry the beef in a pan. Add onion and carrot to the pan. Add the chicken to the pan. Add the tomatoes and stir. Add the potatoes to the pan.

**GT**: Brown 400gm of sliced beef on a hot pan. Fry onions until golden then add garlic carrots and red pepper fry for 5 mins. Now add the beef 2 tbsp of flour 1 tsp of paprika 1 tbsp of tomato puree 2 bay leaves and 300ml beef stock. Add 200 gram canned tomato 100ml red wine sour cream and mix well let it simmer for 1 5 hour. Now add 400gm of baby potato and mix it let it cook for 30 more min.

**MART**: Add flour to a bowl and whisk. Cut the chicken into pieces. Coat the chicken in flour. Coat the chicken in flour egg and breadcrumbs. Fry the chicken in a pan. Drizzle the sauce on top of the bread. Add sauce to the pizza. Bake the dish in the oven.

**COOT (Ours)**: Mix parmesan cheese black pepper and garlic powder. Cover the chicken in the bag. Coat the chicken in the flour. Coat the chicken in the egg and coat with flour. Place the chicken in a pan and fry it on a pan. Pour sauce on top of the chicken and top with mozzarella cheese. Sprinkle parmesan cheese on top. Bake the chicken in the oven.

**GT**: Mix bread crumbs and parmesan cheese. Pound the chicken. Rub salt and pepper onto the chicken. Rub flour onto the chicken dip it in egg and coat with breadcrumbs. Fry the chicken in a pan. Spread sauce over the chicken. Top the chicken with mozzarella cheese. Bake the chicken in the oven.

**MART**: Add tomatoes and beef to a pot. Add water to the pan. Add tomato puree and salt. Add the beef and parsley to the soup. Add the beef to the pot. Add water to the soup and let it simmer. Add the soup to the soup.

**COOT (Ours)**: Add the tomatoes and onions to a food processor and blend them. Add the tomatoes and a bay leaf to the pot. Add the tomatoes and simmer. Remove the tomatoes from the pot and let it cook. Remove the tomatoes from the pot and let it cook. Strain the soup to a boil and let it boil. Turn on the heat and heat to a boil.

**GT**: Add tomato onion green chili and rice to a pan. Add water to the pan. Boil the ingredients and then turn down the heat. Strain the ingredients. Blend the ingredients. Add the water to the mixture and strain. Boil the soup.

**MART**: Add flour to a bowl and whisk. Cut the chicken into pieces. Coat the chicken in flour. Coat the chicken in flour egg and breadcrumbs. Fry the chicken in a pan. Drizzle the sauce on top of the bread. Add sauce to the pizza. Bake the dish in the oven.

**COOT (Ours)**: Mix parmesan cheese black pepper and garlic powder. Cover the chicken in the bag. Coat the chicken in the flour. Coat the chicken in the egg and coat with flour. Place the chicken in a pan and fry it on a pan. Pour sauce on top of the chicken and top with mozzarella cheese. Sprinkle parmesan cheese on top. Bake the chicken in the oven.

**GT**: Mix bread crumbs and parmesan cheese. Pound the chicken. Rub salt and pepper onto the chicken. Rub flour onto the chicken dip it in egg and coat with breadcrumbs. Fry the chicken in a pan. Spread sauce over the chicken. Top the chicken with mozzarella cheese. Bake the chicken in the oven.

Table 12: **Random Captioning samples on ActivityNet (ae-val split).**

**MART**: A man is seen speaking to the camera and leads into him holding up various objects and presenting them to. He then cuts the knife and cuts the sandwich while still speaking to the camera. He then puts the sandwich into the pan and cuts it in half. He then puts the sandwich into the sandwich and puts it in the end.

**COOT (Ours)**: A chef demonstrates how to make a sandwich using bread , then he puts a knife in a kitchen and. Then , the man puts the bread on a bread and cuts it in half. After , the man puts the sandwich in the bread and put it in a plate. Next , the man cuts the bread and put on top of the sandwich.

**GT**: A man shows ingredients for a mortadella sandwich. The man cuts the bred in four pieces and puts mustard and then brown on the stove. Then, the man fries an egg and puts it on the bread as well the mortadella, green leaves, cheese and ketchup. After, the man cuts the sandwich in two and eat one.

**MART**: A person is seen sitting in front of a large pile of grass and holding a stick. The person then puts the tire on the machine and begins putting the tire on.

**COOT (Ours)**: A person is seen using a tool on a machine and piecing together with the camera. The man continues to use the machine on the machine and ends by taking out more out of the machine.

**GT**: A person is seen walking in with a tire on a plank and painting the tire. The person then un does the tire and places the rubber tightly around the side.

**MART**: A small group of people are seen swimming around a pool throwing a ball around to one another. The people continue playing with one another and end by throwing the ball back and fourth.

**COOT (Ours)**: A large group of people are seen swimming around a pool throwing a ball around to one another. The people continue playing with one another and ends with a large group of people watching on the sides.

**GT**: A video of water polo is shown in the gym. A few people watch and the ball goes back and forth.

**MART**: A person is seen sitting in front of a large pile of grass and holding a stick. The person then puts the tire on the machine and begins putting the tire on.

**COOT (Ours)**: A person is seen using a tool on a machine and piecing together with the camera. The man continues to use the machine on the machine and ends by taking out more out of the machine.

**GT**: A person is seen walking in with a tire on a plank and painting the tire. The person then un does the tire and places the rubber tightly around the side.

Table 13: **Random Captioning samples on ActivityNet (ae-test split).**

**MART**: A woman stands on front a house talking. The woman drives the lawn mower with a mower. The woman drives the lawn mower. The woman pushes the lawn mower along the grass. The woman talks to the camera.

**COOT (Ours)**: We see the title on the field , white and white text. We then see a man mowing his lawn. The man stops and talks to the camera. The man stops and turns around. We then see the grass again.

**GT**: The video begins with a picture of a lawn along with a company name and website. The video cuts to a man riding a lawnmower, cutting the grass in a nice neighborhood. When he begins, some kids are playing in the road. At one point, a car passes by. The video ends with the picture of the lawn showing the company name and website.

**MART**: A group of women are dancing on a stage. They are dancing together in a room. They are dancing together.

**COOT (Ours)**: A large group of girls are seen standing together followed by a woman dancing and performing a dance routine. The woman continues speaking to the camera while more people are seen dancing around and leads into a group of. The group continues dancing with one another and ends with a woman speaking to the camera.

**GT**: Several girls are in a classroom dancing and doing ballet. The instructor then comes to talk briefly before continuing on coaching the girls. After,the exercises continue and the girls do leaps and jumps in the room before the outside of the dance studio is shown.

**MART**: People are gathered around a street watching. They are holding flags in their hands. A man in a white shirt is standing next to a fence.

**COOT (Ours)**: A man plays bagpipes while people watch on the sidewalk. A person in a black shirt plays the bagpipes. A person in a white shirt walks past the person.

**GT**: A man on stilts is playing the bag pipes on a street. A bus passes on the street behind the man. A street sign on a pole is shown.

**MART**: A group of women are dancing on a stage. They are dancing together in a room. They are dancing together.

**COOT (Ours)**: A large group of girls are seen standing together followed by a woman dancing and performing a dance routine. The woman continues speaking to the camera while more people are seen dancing around and leads into a group of. The group continues dancing with one another and ends with a woman speaking to the camera.

**GT**: Several girls are in a classroom dancing and doing ballet. The instructor then comes to talk briefly before continuing on coaching the girls. After,the exercises continue and the girls do leaps and jumps in the room before the outside of the dance studio is shown.

Figure 2: **Visualization of the video embedding space with t-SNE on ActivityNet-Captions.** We apply t-SNE to reduce the video embedding space to 2 dimensions and visualize videos by one sample frame.