[Reviews · NeurIPS 2020]

Review 1

Summary and Contributions: This paper studies the problem of video-text representation learning with applications to video-text and text-video retrieval. The proposed method falls into the "late fusion" scheme where an individual stream is adopted for each modality and the resulted embedding outcomes are then aligned in a joint embedding space. The main contributions include a cross-modal cycle consistency loss for video-text fusion enhancement and impressive empirical results and practices.

Strengths: The technical design in this paper is sound. The main framework is a transformer-equipped version of [21], which has already given >10% relative improvement on ANet-Captions compared to the original method in [21]. Intensive adoptations of Transformers and attention modules are used in video/text local, global or local+global contextualized encoding. Experimental results are solid and ablation studies are convincing.

Weaknesses: Overall, the paper has made significant progress towards better video-text retrieval, from both efficiency aspect (compared to "early fusion") and empirical results aspect. However, methodology-wise, this paper is bringing limited excitement as Transformers and attention modules are frequently used for video and text encoding. Some claims are not well supported. For instance, i) [CLS] also adapts attention for temporal aggregation, at a even denser level than the proposed method; ii) the attention-aware feature aggregation is not exactly "new" as it can be regarded as a MLP with GELU for self-attention. But we do note that using standard architecture does not diminish its contribution on boosting model performance (53.7 -> 61.3 on R@1 on ANet-Captions). Also, the idea of applying cycle-consistency loss between video clips and sentences is interesting and effective (57.6 -> 61.3 on the same experiment). The clarity of the paper could be improved; details see the "Clarity" section. Future works on more video-text tasks other than retrieval should be considered as the paper claims to learn general video-text representation.

Correctness: The paper has paid sufficient attention to experiment fairness and its improvements are significant.

Clarity: Overall, the paper is well written. Here are some clarification questions. i) Line 77, what does (1, 1) mean when it comes to Eq. 1? Which positive sample is it referring to? ii) In Fig. 2, where does the Global Context caption come from? Or is it just a proof of concept? If say, please note that in the text. iii) Line 212, this paragraph is quite abrupt. Consider moving it to the supplementary. iv) What's the main difference between HSE in Tab. 1 and HSE in Tab. 2? The numbers do not match (45.6 vs. 44.4). v) In Tab. 1, does replacing CoT indicate an average pooling layer?

Relation to Prior Work: Yes, a comprehensive review on related work is conducted. Recent/relevant existing works are acknowledged and compared against in the experiments.

Reproducibility: Yes

Additional Feedback: Final rating =================== After reading the authors' response, the reviewers agree that the novelties of AF and CoT are limited. CMC is considered novel in the context of video-language retrieval and captioning. However, the overall empirical improvements from the proposed method are rather significant, that said, the somewhat incremental designs are effective across multiple major benchmarks and tasks. To summarize, weighing in the novelty of CMC and model performance, my final rating is leaning towards Accept. Besides, the correct reference on [55] should be: Luowei Zhou, Hamid Palangi, Lei Zhang, Houdong Hu, Jason J. Corso, and Jianfeng Gao. Unified Vision-Language Pre-Training for Image Captioning and VQA. In AAAI, pp. 13041-13049. 2020.


Review 2

Summary and Contributions: The authors proposed a hierarchical transformer model for text-video retrieval and a cycle-consistency loss to better optimize the model, which is closer to optimizing the evaluation metric compared metric learning based losses.

Strengths: 1. The proposed approach is straightforward and intuitive. 2. The experiment results on the multimodal retrieval task are shown to be very effective.

Weaknesses: 1. In the ablation studies, CMC has a much larger benefit when AF or CoT is available. I would like to invite the authors to provide some insights on why these components strengthen each other's benefit. 2. Many hyper-parameter optimization was done. More details about how each hyper-parameter, e.g., activation function selection, optimization algorithms and etc., affect the final performance should be provided. This will provide a comprehensive understanding of the contribution of this paper. 3. Currently all the experiments are done with trains set and test set only. Therefore, the model may be over-fitting the validation set. Some cross-dataset evaluation or test set evaluation, e.g., ActivityNet val2, should be provided.

Correctness: The experiment setting has some flaws as mentioned above.

Clarity: Yes, it is easy to follow.

Relation to Prior Work: Multi-modal Transformer for Video Retrieval, ECCV 2020

Reproducibility: Yes

Additional Feedback: Final rating: I did not find clear indication to show that the results are on ActivityNet val2. Please do make it clearer if I did not miss anything, and please also provide more comprehensive ablation results on hyper-parameters. The idea of cycle-consistency is interesting for me and this is my main motivation for maintain my rating as Marginally above the acceptance threshold. Thanks for the work!


Review 3

Summary and Contributions: This paper introduces a Cooperative hierarchical Transformer for video-text modeling. The introduced transformer incorporates low-level and high-level semantics. A cross-modal cycle-consistency loss is leveraged to build the connection between video and text. The results on ActivityNet-captions, video-paragraph retrieval demonstrate the effectiveness of the introduced model.

Strengths: This paper is well motivated. The inter-level cooperation introduces two levels of cues for video-text modeling. The introduced transformer is simple and could be possibly extended in other applications. The attention-aware feature aggregation method is technically sound and effective. The cross-modality cycle-consistency is innovative and interesting.

Weaknesses: 1. The authors claimed that this is first to introduce an attention-aware feature aggregation module for video-text transformers. However, ViLBERT introduced co-attentional transformer layer for image-text modeling. ActBERT introduced Tangled Transformer for video-text modeling. The authors are suggested adding more discussions. 2. What is the value of $\lambda$ on different architectures and datasets? 3. It seems some of the losses have been studied in Zhang et al. [21]. Can the authors summarize the differences between this paper and [21]? 4. In Table 2, the results significantly outperform previous state-of-the-arts. However, when compared to HSE [21] in Table 1, COOT without AF, CMC, CoT outperforms HSE with a clear margin. How do the authors obtain these improvements?

Correctness: The method is technically sound.

Clarity: This paper is well written and clear.

Relation to Prior Work: The discussion is sufficient.

Reproducibility: Yes

Additional Feedback: Final rating =========== I would like to thank the reviewer for responding my comments. Based on the discussion with AC and peer reviewers, I would like to keep on rating unchanged. The main concern is this paper is to limited novelty of AF and CoT. But I do like to idea of CMC.


Review 4

Summary and Contributions: In this paper, the authors propose a hierarchical transformer-based architecture for video-text retrieval. There are two major contributions, 1) the design of a hierarchical transformer network and 2) the propose of the cycle loss based on sentence-clip retrieval. The proposed approach shows good performance on video-paragraph and clip-sentence retrieval tasks. Ablation studies support the effectiveness of the proposed attention-aware feature aggregation layer, contextual transformer, and the cycle loss.

Strengths: 1. The results of the video retrieval task are good. The ablation studies are sufficient to show the effectiveness of the proposed modules in the paper (i.e., attention-aware feature aggregation layer, contextual transformer, and the cycle loss).

Weaknesses: 1. Although the experiments well support the effectiveness of the proposed attention-aware feature aggregation layer and contextual transformer, it might be necessary to better discuss its superiority over other alternatives. For attention-FA, it looks like conventional attention for me. Could you please expand the discussion in Line 103-107 and explain why these two differences will lead to superior performance? Besides, I wish to clarify whether the CLS in Table 4 refers to the [CLS] output from the T-transformer? As an alternative, it is fairer to compare Attention-FA by replacing it with conventional attention, or an extra T-transformer layer and takes the [CLS] output? Other alternatives to the contextual transformer with local-global fusion could be designed to further validate its effectiveness. 2. The proposed framework is only evaluated on video retrieval tasks which might limit its technical impact. The proposed framework has the potential to work on other video-language tasks such as video grounding, and the extended experiments can better support the method’s effectiveness.

Correctness: The manuscript looks correct.

Clarity: Is it correct that both T-transformer and contextual transformer contain a single transformer layer?

Relation to Prior Work: The references are clear.

Reproducibility: Yes

Additional Feedback: ######################## Final rating ######################## Thank you for your feedback. The additional experiments on other tasks address the concern of "limited impact." My concern is still about the source of the improvement and the ablation study design. Overall, I would like to keep the rating unchanged.

[Author Response · NeurIPS 2020]

**All reviewers.** Thank you for the constructive comments and suggestions. Our work (COOT) aims to exploit long-range
temporal context and leverage hierarchy of semantics to learn joint video-text embeddings. COOT consists of three new
components: an Attention-aware Feature aggregation module (AF), a Contextual Transformer (CoT) and Cross-Modal
Cycle consistency loss (CMC). The proposed method achieves state-of-the-art results on retrieval tasks (Table 2 and
Table 3). We obtained larger improvements on paragraph-video retrieval compared to sentence-clip retrieval (Table
3). This indicates success of our model in capturing long-range semantics, which is the main theme of our paper. We
believe that retrieval task is sufficient for measuring the alignment of semantic representations, but we agree that other
tasks like video captioning may further support our claims. We report the results of video captioning in TabA 1-Left.
Note that due to limited time, our method uses a simple caption generation method based on RNNs while the result of
VideoBERT uses more sophisticated transformer based method. Hence, we expect to get even better improvement over
VideoBERT if we use the same captioning method.

| Video Captioning | | | | | Method | Paragraph $\Longrightarrow$ Video | | Video $\Longrightarrow$ Paragraph | |
|---|---|---|---|---|---|---|---|---|---|
| Method | CIDEr | BLEU-3 | BLEU-3 | ROUGE-L | | R@1 | R@5 | R@1 | R@5 |
| VideoBERT+ | | | | | FSE | 11.5 | 31.0 | 11.0 | 30.6 |
| Transformer | 0.49 | 6.80 | 4.04 | 27.50 | HSE | 32.9 | 62.7 | 32.6 | 63.0 |
| COOT + RNN | **0.67** | **6.91** | **4.43** | **27.80** | COOT | **48.5** | **78.9** | **48.9** | **79.5** |

TabA 1: Video captioning on Youcook2 dataset (Left) and Retrieval on AcitivityNet-captions-val2 (Right).

**Reviewer1.** **Transformers and attention modules are frequently used.** We do not claim novelty for using
transformers or attention for video-text representation learning. The novelty is in the way we handle long-range
interactions between semantics. Our model improves the semantic representation by learning the interactions from
long-range video-text data. Particularly the proposed contextual transformer and the cross-modal cycle consistency
loss contribute to this, as shown by Tab.1 in the paper. **AF vs [CLS] and discussion**. The [CLS] token is one way of
pooling the information of a sequence. Alternatively, one could average representations of all tokens with the same
weight (Kim et al. [25]), which will integrate all the semantics equally. Table 6 of Reimers et al.[A1] shows that
average pooling works better than the [CLS] token. We propose AF (Attention-aware Feature aggregation) to learn
attention weights that emphasize the most relevant content of the input. In Tab.3 of the supplementary material we
have a comparison to [CLS] and average pooling. It shows the benefit of AF. There have been ideas similar to our AF
module in text-only or video-only domains, yet we are the first to apply this idea to the joint video-text domain, which
is an additional but admittedly not groundbreaking contribution. We will discuss this a bit more in the final manuscript.
**i) What means (1,1)?** Only negative pairs are considered, i.e. the D(x,y) parts of positive pairs in Eq.1 become 0. **ii)**
**Fig2. Global context.** It's simply the concatenation of all local captions of the video, i.e., the paragraph.**iii) Line 212**.
Thanks. We will follow your suggestion. **iv) HSE in Tab1 and Tab2.** HSE in Tab2 is copied the HSE original paper.
HSE in Tab1 is our reproduced results of the same architecture to have fair ablations. **v) Replacing CoT.** We use a
one-layer transformer and then average pooling, but we do not use the cross attention layer.

**Reviewer2.** **Why do modules strengthen each other?** We observed that networks with higher performance benefit
more from our CMC loss. This is probably because with more meaningful features the cycle consistency is more
informative. We also believe there is a mutual regularization effect. **Hyper-parameter optimization.** We designed the
architecture from scratch. Thus, it is necessary to set proper hyper-parameter ranges. For instance, the learning rate
range of [0.0005-0.001] results in accuracy range of [59.4%-61.6%] with Adam optimizer on Activitynet retrieval task.
We will provide more results on effect of hyper-parameters range in the supplementary material.
**Results on ActivityNet-val2.** We do use cross-validation by holding out a part of training set for tuning the hyper-
paramters. Moreover, we report results on ActivityNet-val2 in TabA 1-Right. Ours outperforms all baselines including
HSE. Note that similar to HSE, our results on val1 is better than val2 which indicates the difficulty of this split. **Prior**
**work.** Thanks. We will cite it.

**Reviewer3.** **1) AF module.** The purpose of the co-attentional transformer and the tangled transformer is different
from our AF module. We designed AF for pooling, while VilBERT and ActBERT use the [CLS] token. We will add
more discussions to the final paper (also see response to R1 in line 16-21). **2) Value of $\lambda$.** For Activitynet we used
$1\,e-2$ and for Youcook2 $1\,e-3$. **3) Difference to [21].** Our architecture is based on transformers while HSE is based
on GRUs. Additionally, we have new components (AF, CoT and CMC) in our design, which significantly improve
over the base architecture ($\sim 9$ points). **4) Improvement of COOT without AF, CMC, CoT over HSE.** We believe,
this improvement comes from using transformers rather than GRUs. This baseline shows that the HSE idea of using a
hierarchy is well compatible with transformers.

**Reviewer4.** **Attention-FA and [CLS] output.** Conventional attention does not involve any pooling. We either need
average/max pooling or define something like a [CLS] token. Please see the response to R1 in line 16 -23. In Table 4,
CLS method means adding a learnable CLS token to the inputs of all T-Transformers and taking its output value as the
pooled value. We agree that it would be interesting to see how CLS works with more layers. However, it will increase
the complexity of the model. Also, Reimers et al.[A1] showed that average pooling performed better than [CLS].
**Alternatives to contextual transformer?** This is good idea for future work. The second last row of Table 1 can be
seen as an alternative where we replace CoT with a one-layer transformer and average pooling. **Other video-language**
**tasks.** We report the results of video captioning in TabA 1-Left. **Single transformer layer?** We use one layer for
T-transformer and two layers for CoT. We will clarify this in the paper.

[A1]- Nils Reimers and Iryna Gurevych, Sentence-BERT: Sentence Embeddings using Siamese BERT-Networks, EMNLP2019


[Meta-Review · NeurIPS 2020]

The reviews are generally positive: 1 accept (score 7), 2 borderline accept (score 6), and 1 borderline reject (score 5). All reviewers acknowledged strong empirical performance as one of the major strengths of this paper. They also acknowledged the proposed CMC (Cross-Modal Consistency) is novel in the context of video-language retrieval, although the idea of cycle consistency itself is already well-explored. However, the reviewers also pointed out that the technical novelty of this work, especially AF (attention-aware feature aggregation) and CoT (Contextual Transformer), is somewhat limited because Transformer and attention are widely being used for similar tasks. I took a careful reading of the paper another time. The novelty of AF, CoT, CMC all seem limited indeed. But they provide good empirical performance improvement, and these are well supported by convincing experiments and ablation studies. I feel that these could be considered a good contribution to the vision/language community. Therefore, I agree with the reviewers' final recommendation of acceptance.